# Hot Workability of Ultra-Supercritical Rotor Steel Using a 3-D Processing Map Based on the Dynamic Material Model

**DOI:** 10.3390/ma13184118

**Published:** 2020-09-16

**Authors:** Xuewen Chen, Yuqing Du, Tingting Lian, Kexue Du, Tao Huang

**Affiliations:** School of Materials Science and Engineering, Henan University of Science and Technology, 263 Kaiyuan Avenue, Luoyang 471023, China; DuyqStephanie@163.com (Y.D.); lian_tingting@126.com (T.L.); 18437955329@163.com (K.D.); huangtao@haust.edu.cn (T.H.)

**Keywords:** X12CrMoWVNbN10-1-1 alloy steel, three-dimensional (3-D) thermal processing map, finite element, power dissipation, instability coefficient

## Abstract

As a new-type of ultra-supercritical HI-IP rotor steel, X12CrMoWVNbN10-1-1 alloy steel has excellent integrative performance, which can effectively improve the power generation efficiency of the generator set. In this paper, uniaxial thermal compression tests were carried out over a temperature range of 950–1200 °C and strain rates of 0.05–5 s^−1^ with a Gleeble-1500D thermal simulation testing machine. Moreover, based on hot compression experimental data and the theory of processing diagrams, in combination with the dynamic material model, a three-dimensional (3-D) thermal processing map considering the effect of strain was constructed. It was concluded that optimum thermal deformation conditions were as follows: the temperature range of 1150–1200 °C, the strain rate range of 0.05–0.634 s^−1^. Through secondary development of the finite element (FE) software FORGE^®^, three-dimensional thermal processing map data were integrated into finite element software FORGE^®^. The distributions of instability coefficient and power dissipation coefficient were obtained over various strain rates and temperatures of the Ø 8 × 12 mm cylinder specimen by using finite element simulation. It is shown that simulation results are consistent with the microstructure photos. The method proposed in this paper, which integrates the three-dimensional processing map into the finite element software FORGE^®^ (Forge NxT 2.1, Transvalor, Nice, France), can effectively predict the formability of X12CrMoWVNbN10-1-1 alloy steel.

## 1. Introduction

In recent years, global warming has become the most serious ecological crisis to the international community. Carbon dioxide (CO_2_) produced from fossil fuel is one of prime reasons for global warming [1]. For a long time in the future, thermal power generation is still the main way to supply electricity in many countries. In order to enhance the power generation efficiency of thermal power stations and reduce CO_2_ emissions, humans began to research how to raise the efficiency of coal power generation. Advanced ultra-supercritical technology is the most convenient solution for CO_2_ emission reduction in coal power generation. With the enhancement of power generation efficiency, the working temperature and pressure of a steam turbine as well as the capacity of a steam turbine are also increasing. Thermal power units have developed from the initial critical power unit, subcritical power unit (steam pressure 16 MPa, steam temperature 538 °C), and supercritical thermal power unit (steam pressure 24 MPa, steam temperature 566 °C) to the ultra-supercritical power unit (steam pressure 26 MPa, steam temperature 600 °C). The efficiency of power generation increased from 39% to 45% [2]. Therefore, the performance of the materials is very important [3,4] in order to produce a high and medium-pressure rotor, which is a key component of a generator set with high quality. X12CrMoWVNbN10-1-1 alloy steel contains Cr, Mo, Nb and other elements, which results in good corrosion resistance, heat resistance, anti-fatigue performance and lower thermal expansion; therefore it is widely used for the manufacture of supercritical or ultra-supercritical generator rotor [5,6]. The hot processing map is an effective tool for judging the formability of materials and the design of metal forming process. The formability of different regions of materials in the forming process can be analysed and predicted by using hot processing maps, so as to obtain the “safe zone” and “flow instability zone” for the hot forming process, which can optimise the thermal processing performance of the ultra-supercritical rotor. 

According to irreversible thermodynamics theory, physical system simulation and the continuum mechanics of large plastic deformation, Prasad and Gegel constructed a dynamic material modelling-dynamic material model (DMM) [7]. Prasad et al. [8] determined a flow instability criterion of materials through the principle of maximum entropy production rate proposed by Ziegler [9]. In view of the errors and limitations of the instability criterion proposed by Prasad et al., Murty and Rao [10] proposed an instability criterion applicable to all true stress–strain rate relationships in 1997. Based on Lyapounov function, Gegel proposed a criterion for judging the stable flow of materials during plastic deformation [11]. Since the DMM model and instability criterion have been proposed, the thermal processing diagram has been widely used in the field of material formability. Rajamuthamilselvan et al. [12] obtained the best hot working parameters of material by observing and analysing the microstructure of various regions in a hot processing map of 7075 aluminum alloy. Dharmendra et al. [13] constructed a thermal processing map of TX32 magnesium alloy. By observing the microstructure, it was found that the sliding mechanism of the alloy was different in two recrystallization regions of the hot processing map. Jingqi Zhang et al. [14] used the activation energy map, thermal working map as well as a Z parameter map of Ti-15-3 titanium alloy to determine the optimal hot processing window and explored effects of various parameters on the evolution of microstructure. Ravindranadh et al. [15] established a hot working map of Ti-15Al-12Nb alloy on the base of DMM model. It was concluded that there was a certain relationship between the dissipation coefficient and dislocation density through transmission electron microscopy. More importantly, the dislocation density reached a large value when the dissipation coefficient was 22%. Based on the experimental data, H.T. Zhou et al. [16] explained flow stress behaviour with the hyperbolic sine constitutive equation and combined it with the DMM model to establish the hot processing diagram. According to that, the optimum parameters of the AZ80 alloy were obtained. With the in-depth research of hot processing map, numerous material researchers begin to construct a 3-D thermal processing map based on the DMM model to explore deformation characteristics of materials. In order to achieve optimal hot deformation zone, Yu Sun et al. [17] superposed the power dissipation diagram on the instability diagram and then established a 3-D hot working diagram under various conditions of different parameters. Moreover, microstructure evolution was explored to the mechanisms during the process. Juqiang Li [18] drew the 3-D hot working map of AZ80 magnesium alloy composed of temperature, strain and strain rate. As a result, two regions of dynamic recrystallization were determined. Through microstructure analysis, the accuracy of three-dimensional instability maps under different criteria is compared. For the sake of researching the thermal formability of AA7050 aluminium alloy, S. Wang et al. [19] established constitutive equation and a 3-D thermal processing diagram of material to determine the processing conditions for dissipation and instability of AA7050 aluminum alloy during the whole deformation process. A. Mohamadizadeh et al. [20] researched the elevated temperature deformation behaviour of Fe–18Mn–8Al–0.8C steel with temperature ranges of 600–1000 °C and established a 3-D thermal processing diagram of the material containing strain for determining the stability and instability deformation conditions related to microscopic evolution. Yang Liu [21] proposed a constitutive equation including strain compensation, and analysed the thermal deformation behaviour of 6063 aluminium alloy. On the basis of the DMM model, a 3-D hot processing map was generated. As a result, it was concluded that the optimal deformation window of material was in the areas of 673–723 K/0.01–0.1 s^−1^. The accuracy of optimal deformation conditions was verified by using the explicit differential analysis method.

The formability of materials refers to the maximum deformation capacity that can be achieved without coarse grain, mixed crystal and microcracks in the process of plastic deformation [22]. It is an imperative index to characterise the volume forming ability of materials (for instance, extrusion, forging and rolling). The hot processing map reflects the variation of formability with different parameters. However, it can only qualitatively analyse the formability of the workpiece under a specific condition. The deformation behaviour of the workpiece is more complicated during the process. Additionally, the strain, temperature and strain rate in diverse positions of parts are different in deformation mechanisms. Therefore, it is hard to forecast the formability of workpiece correctly only by means of thermal processing maps during hot deformation. For metals with strain softening, stresses rise along with the rise in strain at the beginning of deformation and then will reduce when they reach a maximum value. Furthermore, when the dynamic softening mechanism is sufficient to offset work hardening due to the deformation, the flow stress tends to a certain value. Therefore, the stress varies greatly with the strain. Since traditional hot processing maps do not consider the effect of strain on the thermo-formability of the material, it is not suitable for the metal with strain softening. Moreover, it is found that power dissipation diagrams and instability diagrams vary greatly with the increase in strain for this type of material. Consequently, only 3-D processing maps including strain can accurately describe the formability of materials. It is cognised from preliminary research on hot working maps based on establishment of DMM model to give best processing parameters, which can only qualitatively analyse the formability of materials. At present, there are few studies on the calculation of material dissipation coefficient and rheological instability coefficient in different deformation regions by combining a 3-D hot processing map and finite element analysis method to quantitatively explore the hot forming performance of materials. Considering effect of strain on material deformation, Juan Liu et al. [23] constructed the 3-D hot processing map including strain of AZ31B magnesium alloy. The best hot deformation region of the material is 250–325 °C and 0.1–1s^−1^. Recently, the research on the combination of 3-D hot processing map and finite element (FE) simulation of X12CrMoWVNbN10-1-1 alloy steel for ultra-supercritical rotor is still blank. Furthermore, it is one of the technical bottlenecks for the development of ultra-supercritical generator set to select the best process parameters in the hot working process as well as to effectively control the cracks and microstructure defects. Consequently, it has great practical significance for optimizing thermal working parameters of parts by using computer simulation technology built on the hot processing map. In this paper, on the foundation of the DMM model, the 3-D hot processing map of X12CrMoWVNbN10-1-1 steel is constructed. By integrating the 3-D processing diagram including strain with FE simulation software FORGE^®^, real-time distributions as well as variations of the efficiency of the power dissipation region and instability region are obtained during deformation for the sake of quantitatively analysing the formability of the material.

## 2. Materials and Experimental Procedure

X12CrMoWVNbN10-1-1 alloy steel is widely used in manufacturing HI-IP rotors of ultra-supercritical power generation units. The measured chemical composition of X12CrMoWVNbN10-1-1 alloy steel is shown in Table 1. C is the main element in steel. The hardenability of steel can be improved by increasing the content of C. The Cr element has the function of secondary hardening. The addition of the Cr element to steel can raise the corrosion resistance and quench the ability of this steel. Moreover, the increase in Mo content can also improve the hardenability and prevent temper brittleness. Carbides containing Nb are very stable at high temperatures and can improve the thermal strength of steel. V is a strong carbide forming element. Adding V into the steel can refine the grain size and significantly improve the yield strength of the steel. W is an element that reduces the γ phase region and is a carbide forming element, which can significantly improve the creep strength of heat-resistant steel. A more austenite structure can be obtained by adding Ni; thus, the creep resistance of steel can be significantly improved. Therefore, hardness, strength, toughness, heat resistance, corrosion resistance and wear resistance of the steel can be enhanced by adding some alloy elements.

The material weighing 150 kg was melted in a high-temperature furnace and then molded. Then, the surface defects of as-cast ingots were cleaned up or peeled completely. The ingot is then rounded with a diameter of 100 mm. The sample is a cylinder sample of Ø 8 × 12 mm, which is cut from the as-forged round bar with a diameter of Ø 100 mm by wire cutting technology. Isothermal compression was carried out on the Gleeble-1500D thermal/mechanical simulation machine (Dynamic Systems Inc, New York, NY, USA). Figure 1 is the hot compression process diagram. Before compression, the specimens were heated to a predetermined temperature at the rate of 5 °C/s and kept for 3 min to ensure that internal temperatures of the sample were uniform and stable. In the sake of reducing the friction between specimen and fixture, a layer of lubricant was applied to the interfaces of the cylinder specimen. During the compression process, 30 groups of tests were carried out at 950, 1000, 1050, 1100, 1150, 1200 °C and strain rates of 0.05, 0.1, 0.5, 1, 5 s^−1^. The samples were totally compressed to the true strain of 0.7. After isothermal compression, samples were water quenched immediately. The original austenite grain cannot grow up in time under the rapid cooling state, the microstructure of the material after hot compression is retained. For the sake of observing the microstructure, quenched samples were cut along the axial direction, then ground with a different size of sandpaper (200/400/600/800/1500) and polished with a metallographic polishing machine (MP-2B) (Laizhou Weiyi Experimental Instrument Manufacturing, Yantai, China). Then, 1.25 g of potassium permanganate was mixed with 100 mL 10% dilute sulfuric acid to make the metallographic etching solution. The polished sample was put into the corrosion solution at 75 °C for 15 s until there appeared small bubbles on the surface of sample and the colour changed from silver white to light grey. After the sample was cleaned and dried, the microstructure was observed with an optical microscope (Olympus-pmg3) (Olympus Corporation, Tokyo, Japan). Figure 2 shows the original microstructure of X12CrMoWVNbN10-1-1 alloy steel with an average grain size of 58.13 µm. In order to present the internal deformation of the sample more completely, the metal flow line arrangement experiment was carried out on the deformed sample. The polished samples were put into a 1:1 hydrochloric acid aqueous solution at 65 °C for 15 min. Figure 3 is the transverse macrostructure of upsetting sample. 

## 3. Results and Discussion

### 3.1. Flow Curves of X12CrMoWVNbN10-1-1 Alloy Steel

In Figure 4, stress–strain curves of X12CrMoWVNbN10-1-1 alloy steel are shown after hot compression. It can be illustrated from curves that three parameters (strain rate, temperature and strain) have a remarkable impact on flow stress. In the compression deformation process, stresses decrease when the temperature rises by 950 °C to 1200 °C with the same strain rates. For instance, when the strain rate is 0.5 s^−1^ and the temperature changes from 950 to 1200 °C, peak stress decreases by 174 MPa. At same deformation temperatures, with the rise in strain rate, peak stresses gradually increase. For example, when the temperature is 1100 °C and the strain rate is from 0.05 to 5 s^−1^, peak stresses rise by 69.34 MPa. Furthermore, at the initial deformation state, stresses increase greatly with the rise in strain. The main reason for this phenomenon is the change of flow stress caused by dislocation movement. The raise of the curve is caused by the rapid proliferation of dislocations. The dislocation density continues to increase, resulting in work hardening and increasing stress [24,25]. When the curve reaches the peak value, the curve shows a pronounced downward trend, which indicates that the degree of softening, such as the dynamic recrystallization mechanism and dynamic recovery mechanism, is greater than that of work hardening [26]. The obvious downward trend can be seen with the deformation conditions of a temperature of 1000 °C and a strain rate of 0.1 s^−1^. When the curve reaches the peak value, there is no pronounced downward trend, indicating that the softening effect is in equilibrium with the work hardening effect [27]. For instance, with the deformation condition of 1100 °C and 5 s^−1^, the curve shows a stable state. Consequently, deformation temperature, strain and strain rate are critical factors that affect the flow stress of X12CrMoWVNbN10-1-1 steel during plastic deformation.

### 3.2. Processing Map of X12CrMoWVNbN10-1-1 Steel during the Thermoplastic Deformation Process

#### 3.2.1. Establishment of Power Dissipation Map

Under the condition of constant deformation temperature and strain rate, the thermal deformation behaviour of materials was described by dynamic constitutive model [28,29]:(1)$$\sigma =K{\dot{\varepsilon }}^{m}$$

Here, σ is stress, $\dot{\varepsilon }$ represents strain rates, K is constant, and m represents strain rate sensitivity.

As it is generally considered that the strain rate sensitivity coefficient m is constant at certain strain rate and temperature conditions, the range of m is 0 to 1. Therefore:(2)$$m=\frac{\partial \ln\sigma }{\partial \ln\dot{\varepsilon }}$$

Gegel [7] regards the forging process as a thermodynamic closed system. The energy P input into the system is separated into power dissipation G and the dissipation margin J. The relationship between three parameters can be represented as:(3)$$P=G+J=\int_{0}^{\dot{\varepsilon }}\sigma d\dot{\varepsilon }+\int_{0}^{\sigma }\dot{\varepsilon }d\sigma$$
where P is the total power absorbed per unit volume, G is the total energy consumed when material deforms and J refers to energy consumption by microstructure evolvement during deformation.

Murty et al. [30,31] believe that when the strain rate sensitivity coefficient m is 1, J is up to the maximum value. For instance, the power dissipation coefficient in Formula (4) is a dimensionless parameter, which can be written in Formula (5):(4)$$J_{\max}=\frac{\sigma \dot{\varepsilon }}{2}$$
(5)$$\eta =\frac{J}{J_{\max}}$$

On the basis of Formula (5), the relationship between strain sensitivity coefficient m and the power dissipation coefficient are obtained as follows:(6)$$\eta =\frac{2m}{m+1}$$

The power dissipation diagram is constructed according to reference [32]. With Formula (2) and the flow stress curve of X12CrMoWVNbN10-1-1 alloy steel, the relationship between lnσ and $\ln\dot{\varepsilon }$ can be acquired. Taking the true strain 0.4 and 0.5 as examples, as shown in Figure 5, it can be seen that there is a strong linear relationship between lnσ and $\ln\dot{\varepsilon }$ at different temperatures. The fitting degree is greater than 0.95, which shows that the material conforms to the hypothesis of Formula (2). Therefore, the above formula can be used to calculate the processing map.

For ensuring the accuracy of the data, the cubic spline curve can be used to fit lnσ and $\ln\dot{\varepsilon }$ under the same strain and temperature, which can be illustrated as Formula (7):(7)$$\ln\sigma =a+b\ln\dot{\varepsilon }+c{\left(\ln\dot{\varepsilon }\right)}^{2}+d{\left(\ln\dot{\varepsilon }\right)}^{3}$$

Taking true strain 0.4 and 0.5 as examples from Figure 6 shows, with a fitting degree of each curve above 0.97, that the relationship between lnσ and $\ln\dot{\varepsilon }$ can be described by Formula (7). Thus, Formula (8) is obtained:(8)$$m=b+2c\ln\dot{\varepsilon }+3d{\left(\ln\dot{\varepsilon }\right)}^{2}$$

According to Formulas (7) and (8), strain rate sensitivity coefficient m is calculated. By substituting m values into Formula (6), the power dissipation coefficient can be obtained. As shown in Figure 7, the maximum dissipation coefficient is 0.34 when the strain is 0.2. When the strain is up to 0.5, the maximum value of the dissipation coefficient is 0.49. It can be seen that as the amount of deformation increases, the dissipation coefficient shows an increasing trend.

#### 3.2.2. Establishment of Instability Map

Ziegler [9] proposed the plastic rheological instability of materials and obtained the continuous instability criterion Formula (9) of dynamic material model:(9)$$\xi (\dot{\varepsilon })=\frac{\partial \ln(\frac{m}{m+1})}{\partial \ln\dot{\varepsilon }}+m<0$$

The meaning of Formula (9) is that, when the entropy generation rate of a system is less than the strain rate applied to the system, the plastic rheology will be localised, resulting in rheological instability. According to Formulas (8) and (9), this can be derived as follows:(10)$$\xi (\dot{\varepsilon })=\frac{m^{\prime }}{m(m+1)}+m=\frac{2c+6d\ln\dot{\varepsilon }}{m(m+1)}+m<0$$

In the light of Formula (10), $\xi (\dot{\varepsilon })$ can be obtained under various deformation parameters. At same strain, different strain rates and deformation temperature, the equal line diagram of $\xi (\dot{\varepsilon })$ is drawn. The part less than 0 indicates that the rheological instability occurs in the region, that is, the diagram is the instability diagram. As shown in Figure 8a, the region in gray is instability and the part in white is safe zone. Therefore, in the case of the strain 0.2, the instability zone is 0.05–1.076 s^−1^/950–1132 °C.

#### 3.2.3. Establishment of a 3-D Hot Processing Map

Three-dimensional hot processing maps are different from traditional 2-D hot processing maps. Based on the theory of 2-D machining diagrams, the influence factor of strain is considered. For the material with work hardening and dynamic recovery, the two-dimensional hot processing diagram under a certain strain can reflect the formability of the material. Nevertheless, for the alloy with a pronounced strain softening effect (such as X12CrMoWVNbN10-1-1 alloy steel), three parameters of strain, temperature and strain rate have significant influence on their plastic working process. Therefore, only the 3-D thermal processing diagram considering the strain can accurately describe the forming performance of the material [33,34].

Built on the flow instability diagram and power dissipation diagram at various strains in the above section, the temperature, strain rate and strain are set as three coordinate axes to draw a 3-D power dissipation diagram and 3-D flow instability diagram, as presented in Figure 9 and Figure 10. Figure 9a is a 3-D power dissipation diagram with strain from 0.1 to 0.7. Additionally, the peak value of η rises along with the increase in strain. Figure 9b is a 3-D power dissipation diagram at 950–1200 °C. When the temperature is over 1050 °C, the peak value of η rises along with the increase in temperature. The area with a power dissipation coefficient greater than 25% also expands. At the same time, high power dissipation regions are mainly concentrated in lower strain rate regions.

Figure 10a is a 3-D flow instability diagram with strain ranges of 0.1–0.7. It is shown that the unsafe region expands with the rise in strain. As strain is from 0.1 to 0.3, it is mainly concentrated in a low strain rate and low temperature region (T = 950–1050 °C, $\dot{\varepsilon }$ = 0.05–0.6 s^−1^). When strain is from 0.4 to 0.7, the high-strain-rate region will occur instability. Figure 10b is a 3-D flow instability diagram at temperatures of 950–1200 °C. The flow instability area is large at 950–1100 °C where the material is improper for hot processing. When the temperature is over 1100 °C, the flow instability region reduces. In conclusion, the optimal processing area of the material can be obtained as follows: ε > 0.3, temperature ranges of 1150–1200 °C and strain rates of 0.05–0.634 s^−1^.

### 3.3. Microstructure Analysis of X12CrMoWVNbN10-1-1 Alloy Steel

Figure 11 shows the optical microstructure in a characteristic region of hot compression specimens compressed by 50% on various conditions. Compared with the initial average grain size of 58.13 µm, the average grain size of all deformed microstructures is smaller, which indicates that the grain size decreases with the increase in strain. Figure 11a,b are the microstructures at 1100 °C, 0.05 s^−1^ and 1100 °C, 1 s^−1^. As can be seen from Figure 11a, the grain distribution under this condition is uniform, which indicates the structure is in a stable state. Additionally, the average grain size is 15.53 µm. It can be illustrated from Figure 11b that the original crystal grains are elongated in the horizontal direction. Grain boundaries of deformed grains are gradually curved and arched into a sawtooth shape. Through careful observation, it was found that a great quantity of DRX (Dynamic recrystallization) grains appeared at grain boundaries. The maximum grain size is 27.68 µm and the minimum grain size is 2.89 µm in this structure. It can be concluded that when a lower strain rate is applied, due to the longer deformation time, there is more sufficient time for dynamic recrystallization. The grains gradually nucleate, grow and finally present an equiaxed state. As the strain rate increases, the deformation time is shortened. Therefore, the original grains that have not been consumed by recrystallization remain in the structure. The proportion of recrystallized structure decreases. This is because dislocations accumulate rapidly and stress concentration is difficult to release when plastic deformation occurs at a high strain rate, which inhibits dynamic recrystallization nucleation. At the same time, grain boundary migration is not sufficient and recrystallization growth is restrained when plastic deformation occurs at a high strain rate [25]. Figure 11c,d show the microstructure of 1050 °C and 1150 °C at the same strain rate of 0.1 s^−1^, respectively. It can be observed from Figure 11c that a large number of recrystallized grains appear and the microstructure is obviously refined. The average grain size is 9.58 µm. The original grains are basically replaced by new dynamic recrystallized grains. It can be illustrated from Figure 11d that the average grain size of the dynamic recrystallized grains under this condition is 14.75 µm. Additionally, the structure state is in an equiaxed state. At the same strain rate, the average grain size at 1150 °C is larger than that at 1050 °C. Therefore, as the deformation temperature increases, the grain size of the alloy continues to increase. The mechanism is that the dynamic recrystallization includes two stages: nucleation and growth. With the rise in deformation temperature, the atomic oscillation and diffusion rate of the alloy increase, and the slip, climb and cross slip phase shifts of dislocations are easier than at a low temperature. Moreover, the nucleation rate of dynamic recrystallization increases. At the same time, with the increase in temperature, the migration ability of grain boundaries in the alloy is gradually enhanced. Thus, the strength, hardness and plasticity of the material are improved [35]. In conclusion, strain, strain rate and temperature are important parameters affecting hot deformation behaviour of X12CrMoWVNbN10-1-1 alloy steel.

The three-dimensional thermal processing map is the superposition of the power dissipation map and the instability map of the deformation temperature, strain rate and strain space. In principle, as long as the processing parameters are not in the unsafety area, the processing technology is feasible. In order to optimize the machinability and control the microstructure of the material, it is undoubtedly the best choice to process the material in the dynamic recrystallization region. The temperature and strain rate corresponding to the peak value of power dissipation coefficient in this region are the most suitable processing parameters. Therefore, the three-dimensional hot processing map can reflect the variation of formability with different parameters. The value of power dissipation coefficient on processing maps is related to microstructure evolution mechanism of regions. The microstructure evolution mechanism of materials under various parameters can be identified by the distribution of the power dissipation coefficient on the processing map [18]. However, different microstructure evolution mechanisms may correspond to the same power dissipation coefficient; therefore, the microstructure evolution mechanism identified by processing map still needs to be confirmed by physical experiments. Figure 11c,d are the microstructures at the center of the section under the conditions of 1050 and 1150 °C/0.1 s^−1^, respectively. It can be illustrated from Figure 11c,d that the microstructure presents equiaxed crystal structure, which indicates that complete DRX has occurred. Additionally, power dissipation coefficients are over 30% at these conditions. Dynamic recrystallization is a safe hot deformation mechanism. A higher-power dissipation coefficient in the region of safe hot deformation mechanism represents better formability of hot working [21]. However, at 1050/0.1 s^−1^, cracks appeared in other parts of the section, as shown in Figure 12a. Although the dissipation factor is high under this condition, dynamic recrystallization occurs in some parts of the section, but instability occurs in the section. It is also verified that this condition is in the unsafe zone of the hot processing map. It was observed from Figure 12b that microcracks appear in the cross sections. The cracks propagate along the grain boundary. It shows that, due to the local shear action inside the sample, which leads to the initiation and expansion of cracks, the microstructure is unstable under these deformation conditions. Thus, it is crucial to avoid these flow instability regions in the plastic processing of materials.

### 3.4. Workability Analysis of X12CrMoWVNbN10-1-1 Alloy Steel by Integration of FEM and 3-D Processing Diagrams

The optimum process scheme can be acquired by means of a 3-D thermal processing map. Unfortunately, the deformation behaviour of the material is more complicated during thermal deformation. The heat exchange and friction between the material and the dies make the temperature field and deformation field of the material non-uniform. In the forming process of specific parts, three parameters (strain rate, strain and temperature) in various positions are different. For the sake of preventing defects (for instance, fractures and adiabatic shear bands), it is important to predict the formability of the workpiece during the thermal deformation process to obtain forgings with ideal microstructure and mechanical properties. The FE simulation can predict the distribution of various parameters at different positions of specimens during its hot forming process. It is helpful to know the process quantitatively, intuitively, accurately and comprehensively as well as to adjust the process parameters and optimize the whole process [36,37]. When the cylinder specimen is upset, due to the non-uniformity of deformation, the specimen is roughly divided into three deformation zones according to the degree of deformation, as shown in Figure 13. The area I is a sticky area or a hard-to-deform area. This area is subjected to intense triaxial compressive stress with high hydrostatic stress and small variable shape. The area II at the centre of object is easy to deform, which is still under strong triaxial compressive stress. Because it is a little far away from the contact surface, the hydrostatic pressure is relatively low, which is the most favourable position for slip and development deformation. Zone III near the side surface of the cylinder is the free deformation zone. The deformation in this area is relatively uniform under the stress state of two tension and one compression.

In this section, the 3-D hot processing map acquired in the previous section was integrated into FE simulation software FORGE^®^ to simulate the hot compress process for X12CrMoWVNbN10-1-1 material under different deformation conditions. One half of the cylinder of Ø 8 × 12 mm was used for modelling and the simulation was carried out in the same scheme as the experiment. Figure 14, Figure 15, Figure 16, Figure 17 and Figure 18 show the power dissipation coefficient and flow instability distributions of the simulation. The distributions characteristics of the parameters (power dissipation coefficient and instability coefficient) are obtained by analyzing results in deformation process under various conditions. As can be shown from Figure 13, Figure 14 and Figure 15, with the increase in strain, the power dissipation coefficient increases and the low-power dissipation area decreases at 1050 °C/0.1 s^−1^. Region II is a large deformation area where the total dissipation value is larger than other regions and the dissipation value doesn’t exceed 32%. Prasad pointed out that the rheological instability is usually characterized by flow localization and adiabatic shear bands [7]. Adiabatic shear band is a narrow area with high concentration of internal shear strain. During cylinder compression, the adiabatic shear band usually occurs in direction of 45° angle with central axis. Flow localization refers to the non-violent strain concentration which is manifested as local flow. In uniaxial compression, the region of flow concentration is in the diagonal position. From Figure 15, the unsafe zone extends from the outside to the inside and gradually expands to the area II. The part which was in the safe area gradually turns into the rheological instability region, therefore the rheological instability zone of the sample gradually expands with the deformation. It can be seen in Figure 9 that there are cracks at the grain boundary in the metallographic diagram under this condition, which indicates that evolution of the rheological instability zone of thermal compression deformation is consistent with distribution of rheological instability region analysed theoretically and the performance of the material is not fine under this deformation condition.

As can be obviously observed from Figure 16 and Figure 17 with increase in strain rate, the power dissipation coefficient of zone II obviously decreases, and the low-power dissipation zone gradually increases under the condition of 1050 °C and strain of 0.7. When strain rates are from 0.1 s^−1^ to 0.5 s^−1^, the colour of the destabilized area of area II is darkened, indicating that the destabilized area is increased. As strain rates are from 0.5 s^−1^ to 1 s^−1^, unsafe zone remains unchanged, which basically conforms to the law of hot working diagram. Figure 18 illustrated that power dissipation coefficient increases along with the raise of temperatures under the condition of strain rate of 0.5 s^−1^ and strain of 0.7. The changes in region II and region III are most pronounced.

Under the deformation conditions of 1050 °C/0.1/0.1 s^−1^ and 1050 °C/0.3/0.1 s^−1^, simulation results were compared with microstructure and macrostructure, as presented in Figure 19, Figure 20 and Figure 21. As shown in Figure 19, region I is hard to deform and its region is low-power dissipation zone. The grains in this region are coarse, which hardly change during deformation. The maximum grain size is 43.36 µm, the minimum grain size is 16.10 µm, and the average grain size is 33.43 µm in this area. Region II is large deformation area and high-power dissipation area. It can be observed that there are minimal grains distributed among large grains. The minimum grain size is 2.98 µm and the average grain size is 11.12 µm. It is smaller than the average grain size of region I. The region is prone to occur DRX mechanism due to a high temperature, high strain energy and high dissipation. As illustrated in Figure 20, region I is a low-power dissipation region. The corresponding microstructure is mainly large grains. The average grain size is 35.6 µm. Region II is a high-power dissipation region. The corresponding microstructure shows that there are minimal grains in the boundary of large grains, which indicates that the deformation in this area is large. The average grain size of this region is 12.58 µm, which is three times smaller than the average grain size of region I. It can be seen that the microstructure corresponding to zone III has fine grains distributed on one side and large grains distributed on the other side, which is a free deformation zone. The maximum grain size is 37.32 µm, the minimum grain size is 4.25 µm, and the average grain size is 19.56 µm. Figure 21 shows the distribution of flow line on various state. It can be concluded from Figure 21 that the grains are oriented in the direction of the material flow [38]. It is shown that the simulation results are consistent with the microstructure photos and the proposed method that integrates the 3-D processing map into the finite element software FORGE^®^, is able to predict the formability of X12CrMoWVNbN10-1-1 alloy steel during its hot forging process.

## 4. Conclusions

This study investigated hot workability of X12CrMoWVNbN10-1-1 ultra-supercritical rotor material through compression test implemented by Gleeble 1500D thermal simulator under conditions of deformation temperatures of 950–1200 °C, strain rates of 0.05–5 s^−1^. On the base of experimental data, the flow stress–strain curve of X12CrMoWVNbN10-1-1 alloy steel was obtained. It can be illustrated from the curve that deformation parameters have great influence on the formability of X12CrMoWVNbN10-1-1 steel. Flow stresses fell gradually when temperatures rose from 950 °C to 1200 °C at the specified strain rate. As strain rate increased, peak stress increased gradually from 0.05 to 5 s^−1^ and dynamic recrystallization was the main softening mechanism at the same deformation temperature.

Moreover, based on Prasad’s processing diagram theory, combined with dynamic material model (DMM model) and Ziegler instability criterion, the variation law of power dissipation coefficient and rheological instability region in three space dimensions of strain rate, temperature and strain were researched. The optimum hot forming conditions were determined as follows: strain was greater than 0.3, temperature ranges of 1150–1200 °C, strain rate of 0.05–0.634 s^−1^.

In addition, based on the secondary development technology of finite element software, the 3-D hot working drawing was integrated into the finite element simulation software FORGE^®^. The distributions of power dissipation coefficient and rheological instability coefficient during hot compression deformation were obtained. A method for analyzing the formability of the whole process of material thermal deformation was established. Both the stress state machinability of a particular process and the intrinsic machinability determined by the material were analysed.

## Figures and Tables

**Figure 1 materials-13-04118-f001:**
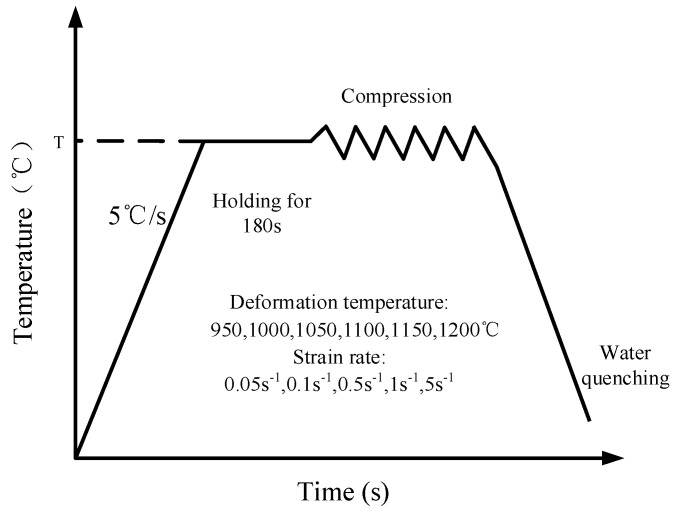
Compression processing.

**Figure 2 materials-13-04118-f002:**
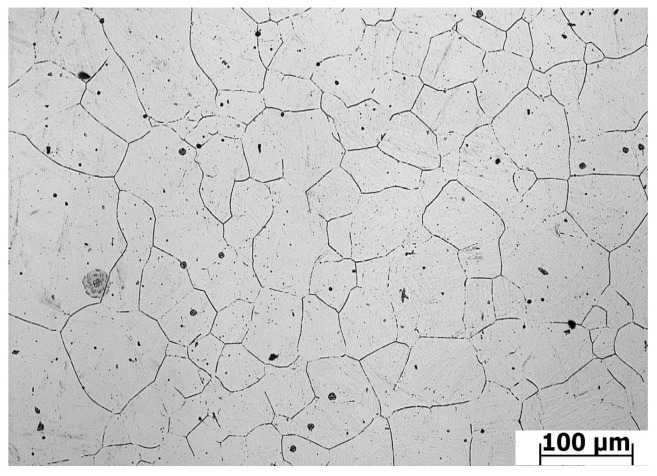
Original microstructure of X12CrMoWVNbN10-1-1 alloy steel.

**Figure 3 materials-13-04118-f003:**
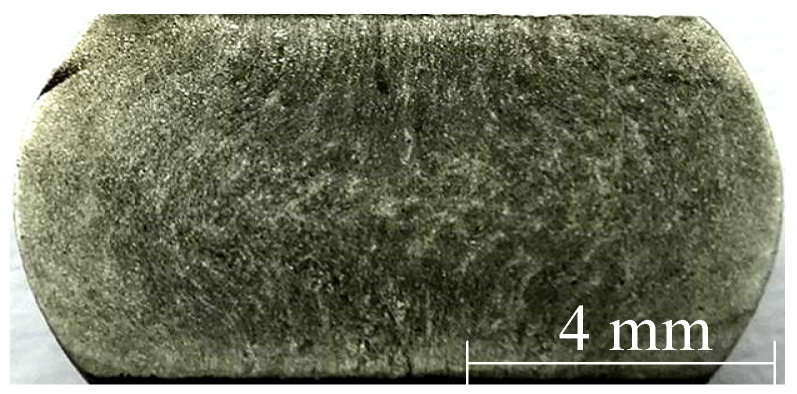
Macrostructure of the upset sample.

**Figure 4 materials-13-04118-f004:**
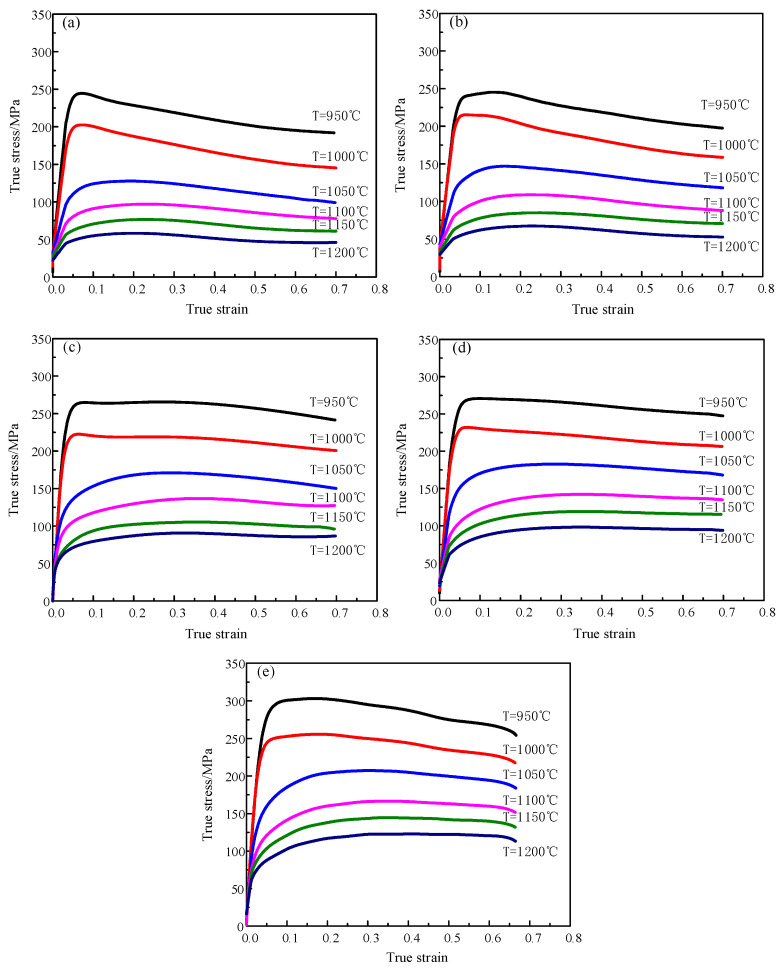
Curves of true stress–strain of X12CrMoWVNbN10-1-1 alloy at different deformation conditions: (**a**) $\dot{\varepsilon }$ = 0.05 s^−1^; (**b**) $\dot{\varepsilon }$ = 0.1 s^−1^; (**c**) $\dot{\varepsilon }$ = 0.5 s^−1^; (**d**) $\dot{\varepsilon }$ = 1 s^−1^; (**e**) $\dot{\varepsilon }$ = 5 s^−1^.

**Figure 5 materials-13-04118-f005:**
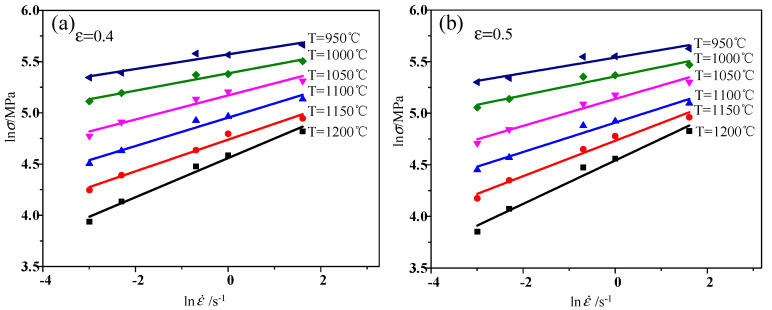
Relationships between lnσ and $\ln\dot{\varepsilon }$: (**a**) *ε* = 0.4; (**b**) *ε* = 0.5.

**Figure 6 materials-13-04118-f006:**
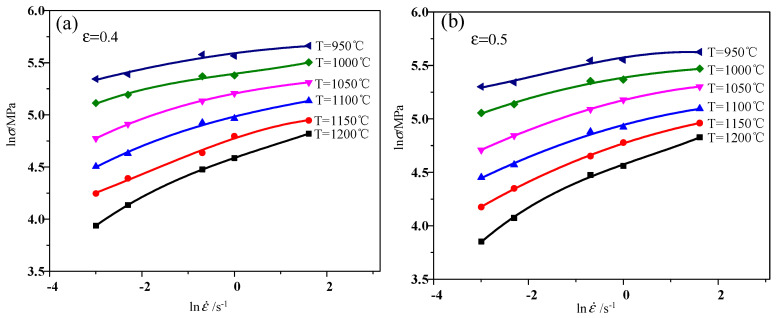
Cubic spline curve fitting of lnσ and $\ln\dot{\varepsilon }$ at different temperatures: (**a**) *ε* = 0.4; (**b**) *ε* = 0.5.

**Figure 7 materials-13-04118-f007:**
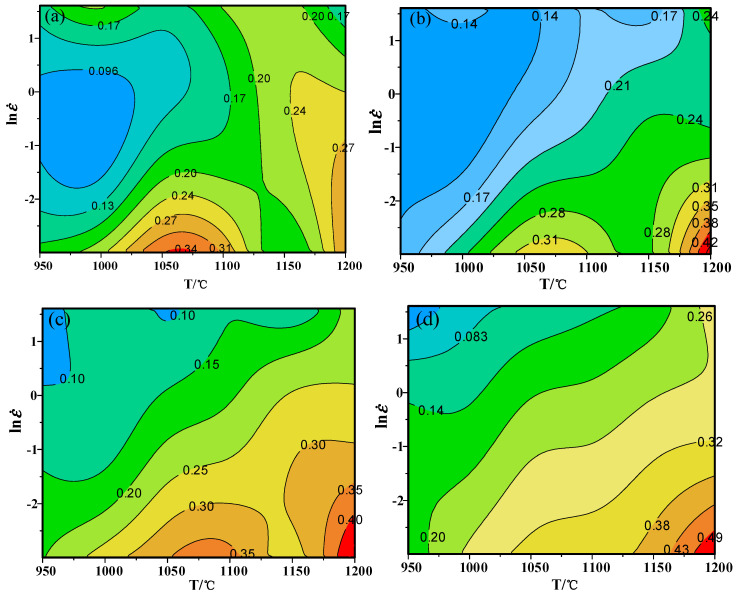
Power dissipation maps at different strains: (**a**) *ε* = 0.2; (**b**) *ε* = 0.3; (**c**) *ε* = 0.4; (**d**) *ε* = 0.5.

**Figure 8 materials-13-04118-f008:**
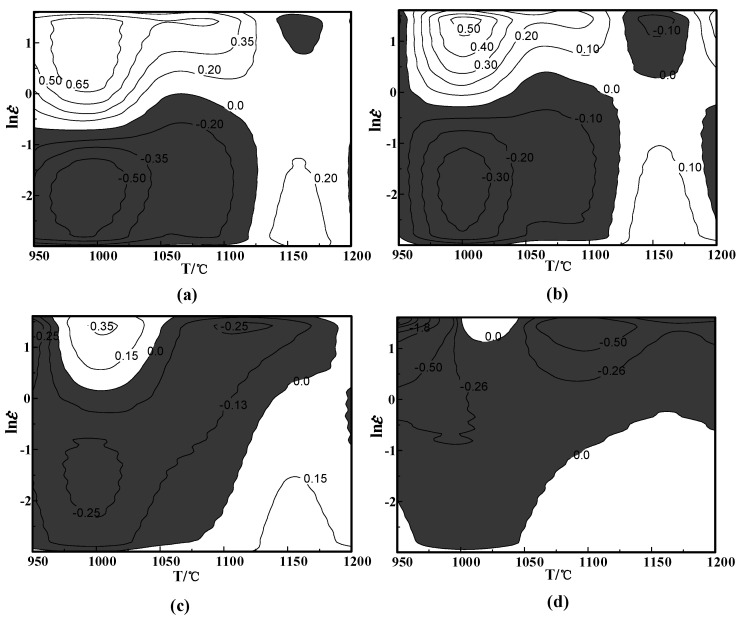
Instability maps at different strains: (**a**) *ε* = 0.2; (**b**) *ε* = 0.3; (**c**) *ε* = 0.4; (**d**) *ε* = 0.5.

**Figure 9 materials-13-04118-f009:**
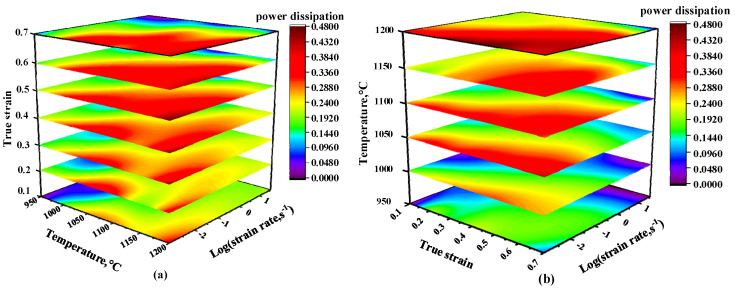
Three-dimensional power dissipation maps at various conditions: (**a**) Three-dimensional power dissipation map at strain ranges of 0.1–0.7; (**b**) The 3-D power dissipation map at 950–1200 °C.

**Figure 10 materials-13-04118-f010:**
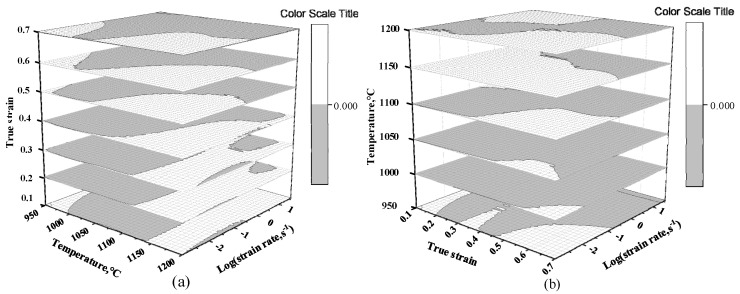
Three-dimensional flow instability maps at various conditions: (**a**) Three-dimensional flow instability map at strain ranges of 0.1–0.7; (**b**) Three-dimensional flow instability map at 950–1200 °C.

**Figure 11 materials-13-04118-f011:**
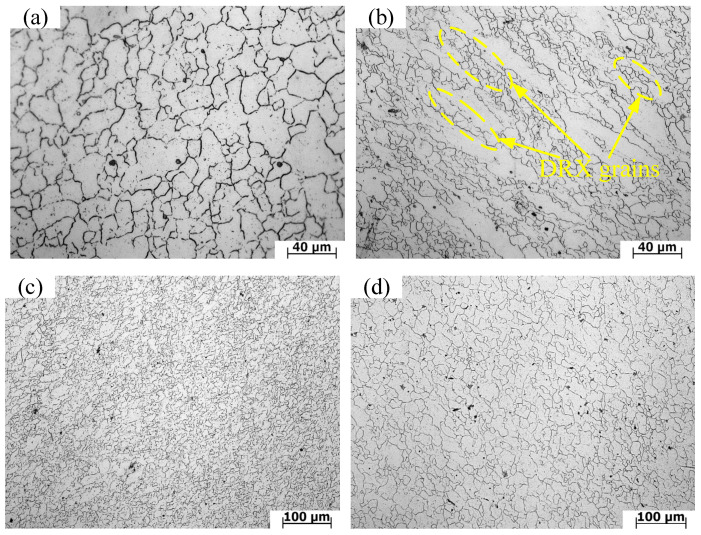
Microstructure of X12CrMoWVNbN10-1-1 alloy steel under various conditions: (**a**) 1100 °C/0.05 s^−1^; (**b**) 1100 °C/1 s^−1^; (**c**) 1050 °C/0.1 s^−1^; (**d**) 1150 °C/0.1 s^−1^.

**Figure 12 materials-13-04118-f012:**
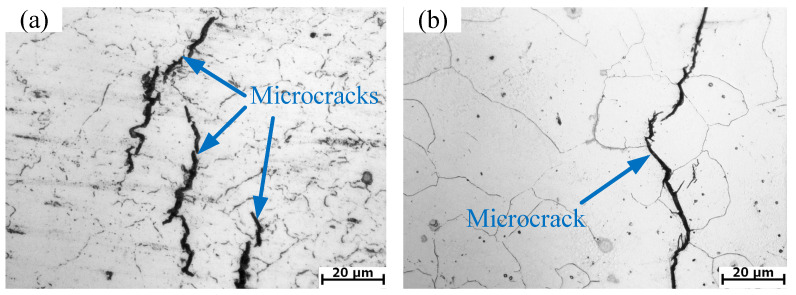
Unstable microstructure of X12CrMoWVNbN10-1-1 alloy steel under various conditions: (**a**) 1050 °C/0.1 s^−1^; (**b**) 1200 °C/5 s^−1^.

**Figure 13 materials-13-04118-f013:**
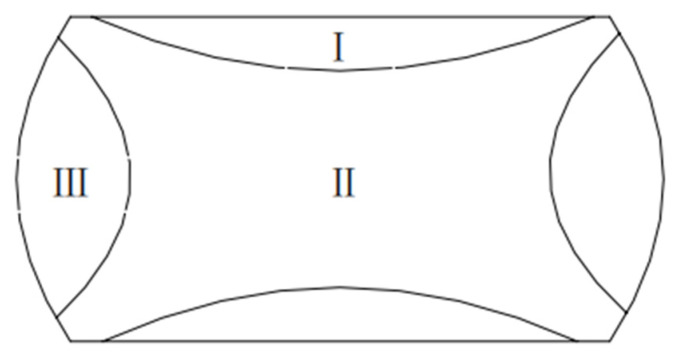
Schematic of the non-uniformity of compressed sample.

**Figure 14 materials-13-04118-f014:**
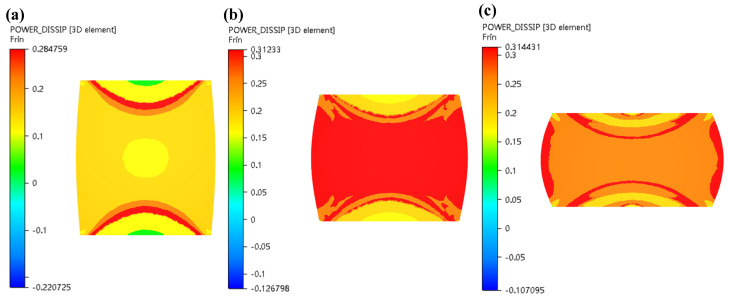
Distributions of η (power dissipation coefficient) in the condition of 1050 °C/0.1 s^−1^ under various strains: (**a**) *ε* = 0.2; (**b**) *ε* = 0.4; (**c**) *ε* = 0.7.

**Figure 15 materials-13-04118-f015:**
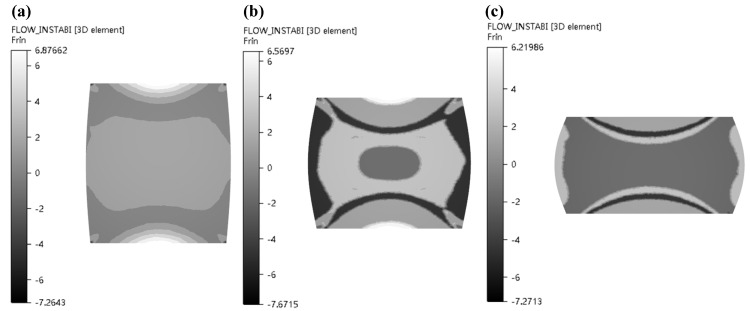
Distributions of ξ (instability coefficient) in the condition of 1050 °C/0.1 s^−1^ under various strains: (**a**) *ε* = 0.2; (**b**) *ε* = 0.4; (**c**) *ε* = 0.7.

**Figure 16 materials-13-04118-f016:**
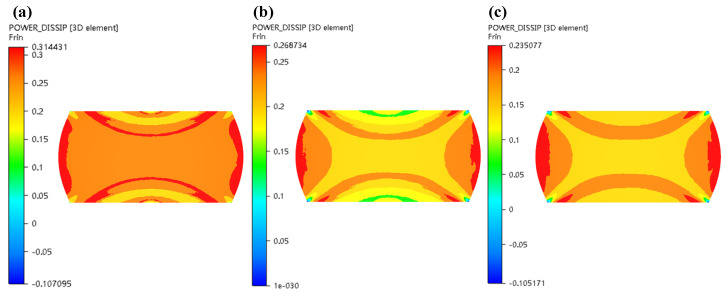
Distributions of η (power dissipation coefficient) in the condition of 1050 °C/0.7 (T = 1050 °C, *ε* = 0.7) under various strain rates: (**a**) $\dot{\varepsilon }$ = 0.1 s^−1^; (**b**) $\dot{\varepsilon }$ = 0.5 s^−1^; (**c**) $\dot{\varepsilon }$ = 1 s^−1^.

**Figure 17 materials-13-04118-f017:**
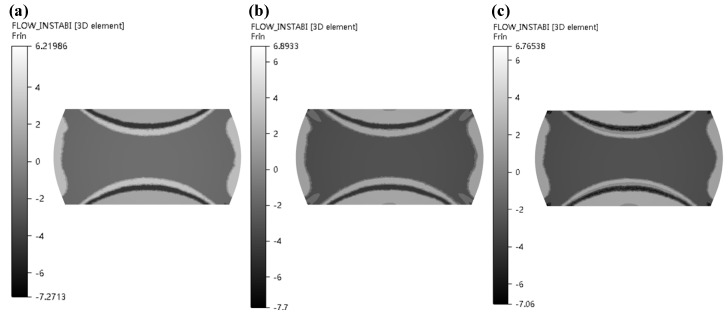
Distributions of ξ (instability coefficient) in the condition of 1050 °C/0.7 (T = 1050 °C, *ε* = 0.7) under various strain rates: (**a**) $\dot{\varepsilon }$ = 0.1 s^−1^; (**b**) $\dot{\varepsilon }$ = 0.5 s^−1^; (**c**) $\dot{\varepsilon }$ = 1 s^−1^.

**Figure 18 materials-13-04118-f018:**
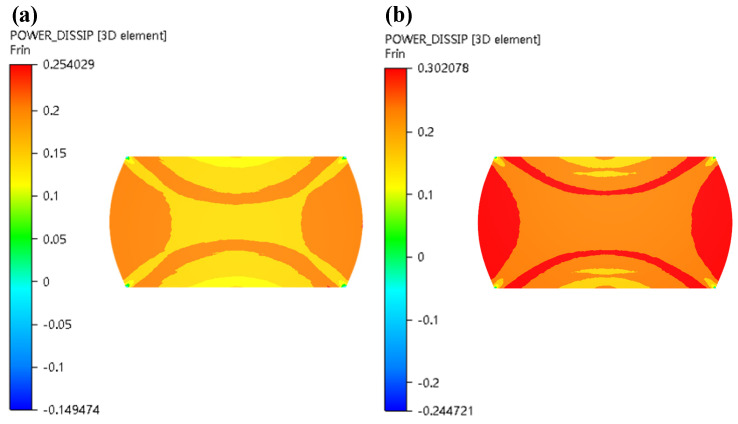
Distributions of η (power dissipation coefficient) in the condition of 0.5 s^−1^/0.7 ($\dot{\varepsilon }$ = 0.5 s^−1^, *ε* = 0.7) under various temperatures: (**a**) T = 1000 °C; (**b**) T = 1100 °C.

**Figure 19 materials-13-04118-f019:**
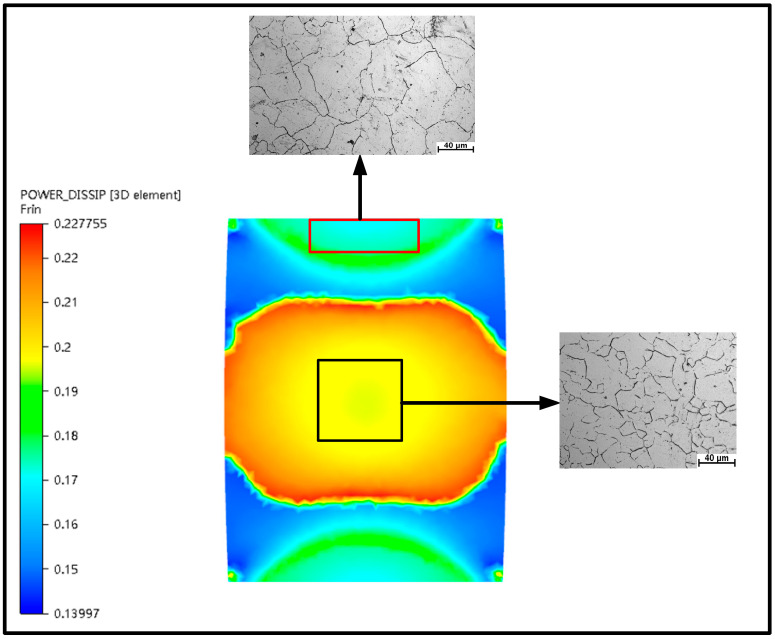
Characteristic diagram of thermal deformation power dissipation at 1050 °C/0.1/0.1 s^−1^.

**Figure 20 materials-13-04118-f020:**
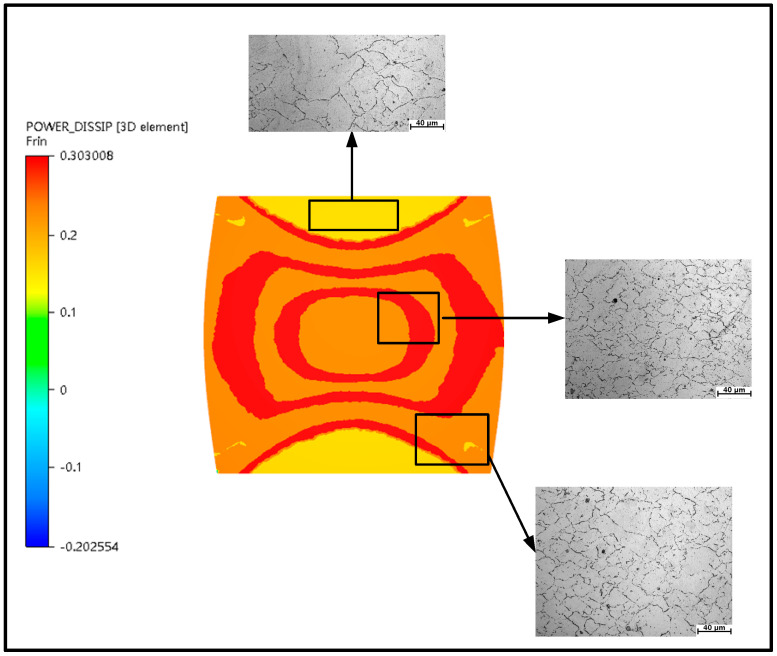
Characteristic diagram of thermal deformation power dissipation at 1050 °C/0.3/0.1 s^−1^.

**Figure 21 materials-13-04118-f021:**
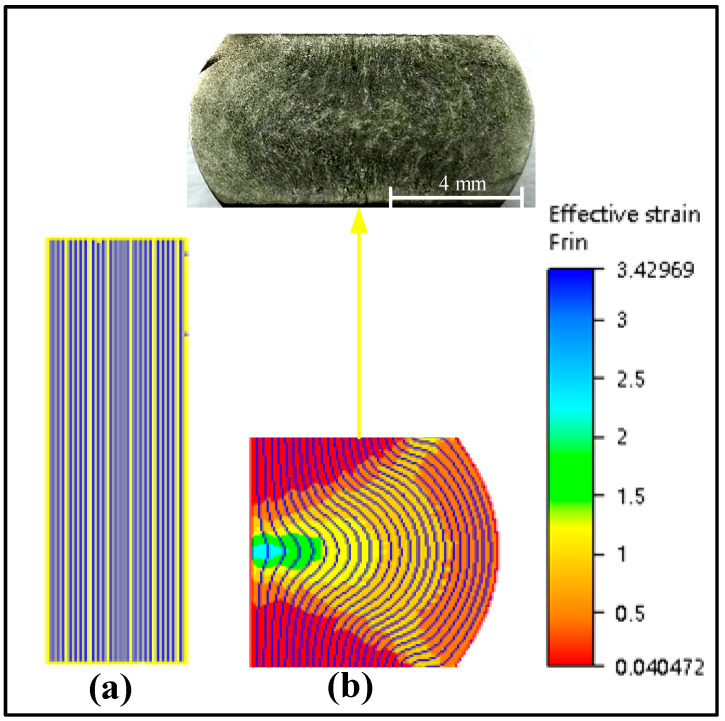
Distribution of flow line on various state: (**a**) start of the process; (**b**) 1100 °C/0.7/0.05 s^−1^.

**Table 1 materials-13-04118-t001:** Measured chemical compositions of X12CrMoWVNbN10-1-1 alloy steel (mass fraction, %).

C	Cr	Mo	Nb	V	W	Ni	Mn	N
0.118	11.00	1.029	0.069	0.207	0.95	0.744	0.420	0.055

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
