# Peer review of "Hot Workability of Ultra-Supercritical Rotor Steel Using a 3-D Processing Map Based on the Dynamic Material Model"

_materials, 2020, doi:10.3390/ma13184118_

Round 1
Reviewer 1 Report
The paper investigates the formability of high Cr ferritic heat-resistant steel. The article compares the simulation and the experimental results, which is an interesting attempt and worth to being published. The optimisation of the processing parameters is the up-to-date problem, and the idea of employing the 3-D processing map seems worth publishing. However, the paper needs some improvements and explanations to specific issues addressed bellow.
- The microstructure analysis should be supported by the quantitative analysis. The size, arrangement, ferret diameters should be estimated. Then the grains measured in specific sample areas should be compared. The discussion of the results should explain how the processing parameters effect on microstructure / macrostructure development and how it connects with the steel formability.
- The flow lines arrangement in upsetting samples macrostructure is directly connected with the strain rate coefficients, which is supported by the literature: https://doi.org/10.3390/ma13092022 and 10.1016/j.jallcom.2017.06.065, In my opinion, the experimental part of your paper needs presenting etched macro-structure of upsetting sample cross-section. Then the work and research material will be completed and the authors' statement ("It is shown that simulation results are consistent with the microstructure photos.") will be justified.
- Please discuss in the paper if your "optimal parameters" do not charm the microstructure and mechanical properties of the final product. Please explain how the process temperature (treatment parameters) affects the microstructure and mechanical properties of the final product. Many works pointed out the microstructural changes due to process treatment parameters, e.g. 10.2355/isijinternational.54.1705
- L133-139 - this phrase is too general. Especially: Therefore, hardness, strength, toughness, heat resistance, corrosion resistance and wear
138 resistance of the steel can be enhanced by adding some alloy elements. - you can explain the meaning of each alloying element. These elements have much more advanced effects than those listed by the authors (only for Cr and Mo). - Table 1 - please explain if it is a nominal chemical composition, or was it measured by the authors?
- What was the initial state of the samples? Heat-treated or as-received without any processing? Were they cut from the rod? The initial macrostructure affects the formability of the steel.
- Please present the same Y-axis range in each graph (fig. 2).
- In the 3.3 section and fig. 9, please clearly describe the obtained microstructure. It can be useful for the potential readers that are not familiar enough with the metallography. You can use the literature, e.g. https://doi.org/10.1016/j.jmrt.2015.06.001
- Authors use X12 name for investigated steel. Please use X12CrMoWVNbN10-1-1 uniformly in the whole text.
- I think that the comparison of macroscopic cross-sections with the simulation results should be presented in 3.4 section.
- Please discuss whether your "optimal formability parameters" are in agreement with the literature? There are many papers relating to the X12CRMOWvNBN10-1-1 steel treatment and formability please discuss your "optimal" temp. values with the literature.
- There are "missinig spaces" and typos that must be improved e.g.: dioxide(CO2); element(FE); ti-15al-12nb; Time(s); X12CRMOWvNBN10-1-1
Reviewer 2 Report
The paper discussed and approach combining 3-D processing map and simulation through FEM developed by means of the software FORGE® in order to predict the formability of X12CrMoWVNbN10-1-1 alloy steel, which is used in the power generation field.
The topic of the paper is interesting and fully aligned to the aims and scope of the journal.
The paper is overall well written and nice to read. The proposed integrated approach is convincing from the technical point of view and the investigation methodology is credible and sound.
The introductory section provides a good overview of the state of the art and correctly frames the proposed research work in the context of the international literature by also relating it with the practical applications.
The pursued experimental procedure is clearly although synthetically described in Section 2. As far as the material is concerned, the chemical composition as well as some discussion on the role of the microalloying elements is provided. However, in order to have a complete and comprehensive description, some notes should also be added on the industrial process through which such material is produced. Finally, as far as the preparation of the samples is concerned, some more details should be added related to the optical analysis.
The obtained experimental results are extensively described in Section 3 by also exploiting the support of well prepared and useful figures. I really appreciate this part. My only suggestion are to avoid the division of figures in two pages and to improve Figure 8, as the choice of the greyscale makes the interpretation of this figure somehow difficult.
The discussed conclusions are well supported by the provided data. However, I suggest to avoid the itemized format and to propose it in a more discursive form. Moreover, the final sentence should be reformulated as follows: “Both the stress state machinability of a particular process and the intrinsic machinability determined by the material were analysed.”
As a further minor formal remark, the use of acronym should be avoided in the abstract.
Round 2
Reviewer 1 Report
I accept all of the authors' responses. The paper was thoroughly improved and can be published in the current form.
Good luck in your future works.